# Mitigating Modality and Language-Style Gaps for Zero-Shot Video Moment Retrieval

## Abstract

Zero-shot video moment retrieval (ZMR) aims to overcome the limitations of traditional approaches that require large-scale datasets annotated with text and its relevant temporal spans. Despite advances in pre-trained vision–language models (VLMs) and multimodal large language models (MLLMs), existing ZMR methods still heavily depend on query-to-context similarity, making them vulnerable to modality and language-style gaps. These gaps lead to unreliable span proposals and unstable moment retrieval results. To address this issue, we propose Self-Similarity-based Moment proposal and Scoring (Self-SiMS) that instead exploits intrinsic relationships within videos, enabling consistent candidate generation and scoring. By deriving self-similarity only from the video content, we circumvent the noisy and mismatched patterns of query–frame or query–caption similarities, thereby mitigating both modality and language-style gaps. Furthermore, we introduce a query-aware MLLM-based reasoning stage to further sharpen alignment between text and video by mitigating modality and language-style gaps. Extensive experiments demonstrate that Self-SiMS achieves the state-of-the-art performance across multiple ZMR benchmarks.

## 1 Introduction

With the rapid expansion of multimedia content, video has become a primary medium for obtaining information. Consequently, retrieving specific moments from videos relevant to natural language queries has garnered significant attention across both academia and industry. A central task in this area is video moment retrieval (VMR) Barrett et al. (2015); Gao et al. (2017); Lei et al. (2021); Xu et al. (2024a), which aims to localize the most relevant temporal segment in a video based on a given text query. This task remains particularly challenging due to the need to pinpoint precise moments within videos that often contain multiple semantically similar scenes.

Previous studies have primarily focused on training VMR models using datasets annotated with temporal spans. However, such approaches require extensive manual labeling, which is both costly and labor-intensive. To overcome this limitation, recent research has explored zero-shot video moment retrieval (ZMR) Radford et al. (2021); Nam et al. (2021); Wang et al. (2022); Zheng et al. (2024); Xu et al. (2025), leveraging vision-language models (VLMs) Li et al. (2022; 2023a) and multimodal large language models (MLLMs) Touvron et al. (2023a;b); Grattafiori et al. (2024) pretrained on large-scale data to perform retrieval without additional supervision. Although these methods have shown promising performance in ZMR, they still face a critical limitation stemming from the unreliable query-based similarity score used to generate and score candidate spans (i.e., moment proposals).

Specifically, existing methods Zheng et al. (2024); Xu et al. (2025) typically rely on the similarity between text queries and video contexts, represented by visual features or MLLM-generated captions. However, these similarities between text queries and video contexts are inherently affected by modality and language-style gaps. As illustrated in Fig. 1(a), method Zheng et al. (2024) that relies on query–frame similarity faces a modality gap, since text queries (stars) and frame-level visual features (blue dots) belong to different modalities. Meanwhile, method Xu et al. (2025) that leverages query–caption similarity still suffers from a language-style gap between human-written queries and MLLM-generated captions (green dots). Under these gap-affected approaches, computing similarities between the query and frame features yields flat or noisy frame-level signals, as

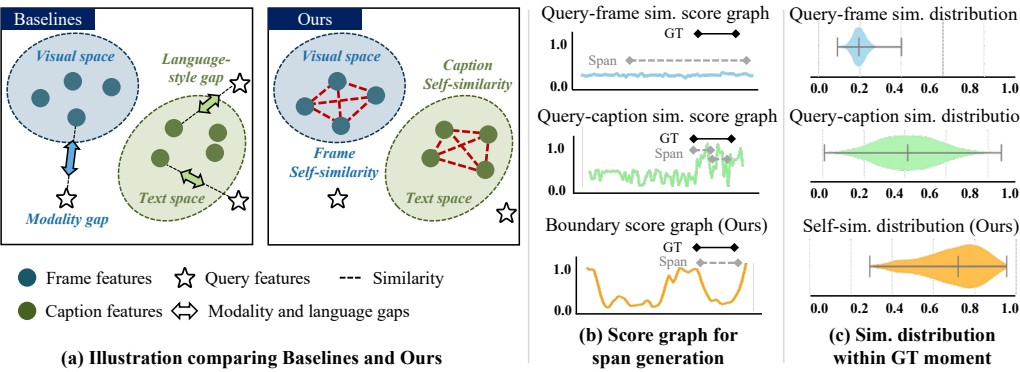

Figure 1: Illustration of modality and language-style gaps in zero-shot video moment retrieval. (a) Baselines (left) depend on query–frame or query–caption similarities, which suffer from modality and language-style gaps, while Ours (right) leverages intra-video self-similarities (SS) among frames and captions to avoid them. (b) Span score graphs from a video instance. Top: Query–frame similarity (QF) is flat across ground-truth and non-ground-truth regions due to the modality gap. Middle: Query–caption similarity (QC) is noisy because of the language-style gap. Bottom: Self-similarity–based boundary scores (Ours) align well with the ground truth. (c) Similarity distributions within ground-truth spans on QVHighlights. Top: QF is low and narrowly concentrated. Middle: QC spreads widely due to noisy curves. Bottom: SS yields consistently higher and more stable values, demonstrating robust and reliable span generation.

shown in the top and middle of Fig. 1(b) which present similarity score graphs for a video instance. The top graph depicts the query–visual frame feature similarity, where the modality gap produces a nearly flat trajectory, offering insufficient contrast to align with the ground-truth span. The second graph shows the query–caption similarity, which alleviates the modality gap but still suffers from the language-style gap; as a result, the similarity curve becomes unstable and fails to provide reliable span boundaries. While these issues are clear at the instance level, similar patterns can also be observed at the dataset level. As illustrated in Fig. 1(c), the query–frame similarity within GT moments are concentrated around low values, despite the fact that high similarity should be observed in these regions, highlighting the distortion caused by the modality gap. In contrast, the query-caption similarity distribution is broadly dispersed, reflecting unstable alignment between queries and generated captions. Thus, existing methods rely on query–frame or query–caption similarities, which are prone to distortion from modality and language-style gaps.

In this paper, we propose Self-Similarity based Moment proposal and Scoring (Self-SiMS), which exploits self-similarity within a video to both generate and score candidate spans. As illustrated on the right of Fig. 1(a), we eliminate the dependence on query–frame or query–caption similarities and instead rely solely on intra-video self-similarities. This design makes Self-SiMS inherently robust to both modality and language-style gaps. This robustness, in turn, enables self-similarity to produce boundary scores that peak at frames marking scene transitions, leading to more reliable localization. As shown at the bottom of Fig. 1(b), these scores consistently align with the ground-truth moments, demonstrating that self-similarity provides cues for detecting event boundaries within a video. Furthermore, the distribution in Fig. 1(c) shows that self-similarity provides stable and elevated scores within ground-truth moments across the dataset, highlighting its ability to capture intrinsic video structure and generate reliable span candidates.

In addition to generating reliable candidates, we introduce span scoring with self-similarity awareness. First, each candidate span is scored based on its top-ranked frame-level query-caption scores. This strategy is designed to avoid the unstable score distributions found in prior work Xu et al. (2025), which result from relying on query-caption scores. These span scores are then further refined by integrating the self-similarity distribution. This refinement step enhances the robustness of scoring by leveraging the internal content consistency of each span to mitigate the impact of modality and language-style gaps. Finally, to accurately select the final span from semantically similar span candidates, we introduce a query-aware MLLM-based re-ranking step. In this stage, representative frames from each span are directly evaluated against the query using an MLLM with a yes/no prompt. Instead of computing similarity after independently generating representations for the query

and video, we leverage the MLLM's query-aware capabilities to further mitigate both modality and language-style gaps, resulting in more accurate span scoring.

To sum up, our contributions are as follows:

- We first explicitly identify and address the modality and language-style gaps in zero-shot video moment retrieval (ZMR), where mismatches between text queries and video contexts, whether visual features or MLLM-generated captions, can lead to inaccurate span scoring.

- To mitigate these gaps, we propose Self-Similarity based Moment proposal and Scoring (Self-SiMS), which leverages self-similarity within video content to derive boundary scores for generating candidate spans and scoring spans based on intra-video relations.

- We further introduce a query-aware MLLM re-ranking step, where candidate spans are directly evaluated against the text query using an MLLM to mitigate the gaps above.

- Our method achieves the state-of-the-art performance on ZMR benchmarks, demonstrating the effectiveness of combining Self-SiMS with query-aware MLLM re-ranking in mitigating modality and language-style gaps and improving ZMR.

## 2 RELATED WORKS

**Video moment retrieval**  Video moment retrieval (VMR) is a fundamental yet challenging task that aims to retrieve relevant temporal segments from an untrimmed video given a natural language query. Early work formulated this as a query-to-segment matching problem using joint video-text feature learning and alignment techniques Barrett et al. (2015); Gao et al. (2017); Krishna et al. (2017). With the introduction of large-scale datasets such as ActivityNet Caba Heilbron et al. (2015) and QVHighlights Lei et al. (2021), subsequent methods explored attention-based models, cross-modal transformers, and contrastive learning to improve segment localization Lei et al. (2021); Sun et al. (2023b); Xu et al. (2024a;b); Li et al. (2022); Ma et al. (2023). Recent approaches further leverage pre-trained vision-language models (VLMs) like BLIP Li et al. (2022), BLIP-2 Li et al. (2023a), and instruction-tuned models Zhang et al. (2023); Maaz et al. (2023) achieving promising results in moment retrieval and highlight detection. Nevertheless, these advances depend heavily on large-scale span annotations, which are costly and time-consuming to obtain, motivating research into weakly-supervised and zero-shot retrieval paradigms.

**Zero-shot video moment retrieval**  Traditional VMR approaches rely on densely annotated datasets that precisely align queries with video moments Lei et al. (2021); Sun et al. (2023b), but such annotations are costly and labor-intensive, motivating research into zero-shot video moment retrieval (ZMR) Nam et al. (2021); Wang et al. (2022). Early zero-shot methods leveraged pretrained VLMs such as CLIP Radford et al. (2021) to compute query–frame similarity without training data Nam et al. (2021), yet they struggled to capture temporal relationships. Moreover, recent studies show that off-the-shelf multimodal large language models (MLLMs) can achieve competitive zero-shot performance without additional training Diwan et al. (2023); Luo et al. (2024); Wattasseril et al. (2023); Huang et al. (2024). Despite these advances, existing methods still rely on query–context similarity using either visual features or MLLM-generated captions for candidate span generation and scoring, which makes them vulnerable to modality and language-style gaps and limits precise moment localization.

**Modality and language-style gaps in text-visual alignment**  Pretrained VLMs Radford et al. (2021); Li et al. (2022; 2023a) trained on large-scale text–image pairs have advanced cross-modal understanding, yet a modality gap remains, causing inaccurate similarity matching. Prior works address this by introducing learnable gap parameters Xiao et al. (2025), enhancing semantic modeling Liu et al. (2024), or generating captions with LLMs to bridge text and visuals. However, LLM-generated captions create a language-style gap with human queries, which some studies mitigate by augmenting queries with complementary descriptions Liu et al. (2023); Xu et al. (2025). Despite these efforts, ZMR still suffers from both gaps, often leading to unreliable span generation and scoring. We propose a framework that leverages video-intrinsic self-relationships to mitigate them jointly.

## 3 METHOD

### 3.1 OVERVIEW

Given an untrimmed video $V = \{v_i\}_{i=1}^L$ consisting of $L$ frames and a textual query $Q$, the goal of the zero-shot video moment retrieval task (ZMR) is to identify temporal span within a video that are semantically aligned with the query, without requiring any training. The output spans are denoted as $T = \{(t_j^s, t_j^e)\}_{j=1}^{N_t}$, where each pair $(t_j^s, t_j^e)$ represents the start and end timestamps of the $j$-th relevant moment, and $N_t$ is the total number of output spans. The overview of our framework is illustrated in Fig. 2. First, we extract the frame features $F^f \in \mathbb{R}^{L \times D^f}$ from the pre-trained visual encoder Caron et al. (2021), where $D^f$ denotes the dimensionality of the frame features. To extract the frame-wise caption features $F^c \in \mathbb{R}^{L \times D^c}$, we employ MLLM Touvron et al. (2023b) to generate captions that describe each frame and a text encoder Reimers & Gurevych (2019), where $D^c$ represents the dimensionality of the caption features. Then, the Temporal Self-similarity Matrix (TSM) of the frames and captions, $M^f, M^c \in \mathbb{R}^{L \times L}$, are derived from $F^f$ and $F^c$, respectively. To obtain candidate spans, we merge TSMs into a single TSM, $M$, and measure the boundary score of each frame using a contrastive kernel, $K$. These candidate spans are scored by the Query-Matching Span Score $S^Q$, where the top-$k_S$ query–caption scores within the span are averaged. Furthermore, the Self-Matching Span Score $S^S$ computed on TSM is added to reflect the context consistency within the span. Finally, $S^Q$ and $S^S$ are integrated into the overall score $S$, which is then used to rank the candidate spans. Finally, we exploit the same MLLM used for generating captions to re-rank top-$k_C$ candidate spans. Specifically, we provide the query and frame pairs to the MLLM and make it answer with 'Yes' or 'No', using the logit of 'Yes' as the re-ranking score.

### 3.2 SELF-SIMILARITY-BASED SPAN GENERATION

**Span Generation with Self-Similairty**  Prior methods generate candidate spans by relying on query-based similarities. This approach, however, is susceptible to critical gaps: a modality gap for direct query-video comparison, and a language-style gap for query-caption matching, both leading to unstable scores. In contrast, our method generates spans from a video's internal self-similarity by identifying content boundaries with a contrastive kernel, making it inherently robust to these gaps. Further details are provided below.

**Span Boundary Scores**  Inspired by the event boundary detection method proposed in Kang et al. (2022), we calculate self-similarity to construct the Temporal Self-similarity Matrix (TSM), $M \in \mathbb{R}^{L \times L}$. First, we apply $\ell_2$-normalization to the extracted features, $F^f$ for frames and $F^c$ for captions, resulting in $\hat{F}^f$ and $\hat{F}^c$, respectively. Then, The TSMs for the frames and captions are computed as $M^f = \hat{F}^f(\hat{F}^f)^\top$ and $M^c = \hat{F}^c(\hat{F}^c)^\top$. To integrate both modalities, we construct the unified TSM $M$ by equally weighting $M^f$ and $M^c$, i.e., $M = \frac{1}{2}(M^f + M^c)$. To evaluate the likelihood of each frame being the boundary of span, we adopt a contrastive kernel $K \in \mathbb{R}^{N_K \times N_K}$ convolved along the diagonal elements to highlight local dissimilarities, where $N_K$ is the size of kernel. The contrastive kernel consists of positive weights in the top-left and bottom-right quadrants, negative weights in the top-right and bottom-left quadrants, and zeros in the entire central cross, namely the middle row and the middle column. A boundary score of the $i$-th frame is calculated as follows:

$$b_i = P_i \odot K, \tag{1}$$

where $P_i$ is an $N_K \times N_K$ patch of $M$ centered on the diagonal element at $(i, i)$, i.e., the submatrix whose rows and columns range from $i - r$ to $i + r$, with $r = \frac{k-1}{2}$. Note that $\odot$ denotes the element-wise multiplication. The TSM is zero-padded to ensure uniform kernel operation across every position. Intuitively, the contrastive kernel assigns a high boundary score to a frame that marks a transition between two spans, where the preceding and succeeding frames are each internally consistent but mutually distinct.

**Candidate Spans**  To generate candidate spans from the boundary scores, we first define a set of instance-wise dynamic thresholds $\mathcal{T} = \{\tau_1, \tau_2, \ldots, \tau_{N_\mathcal{T}}\}$, which adapt to the variability of the frame features $F^f$. The detailed definition of these thresholds is provided in the Appendix. Then, for each $\tau_j \in \mathcal{T}$, we construct boundary set $\mathcal{B}_j$:

$$\mathcal{B}_j = \{\, i \mid b_i \geq \tau_j, \ b_i \geq b_{i-1}, \ b_i \geq b_{i+1} \,\} \cup \{1, L\}, \tag{2}$$

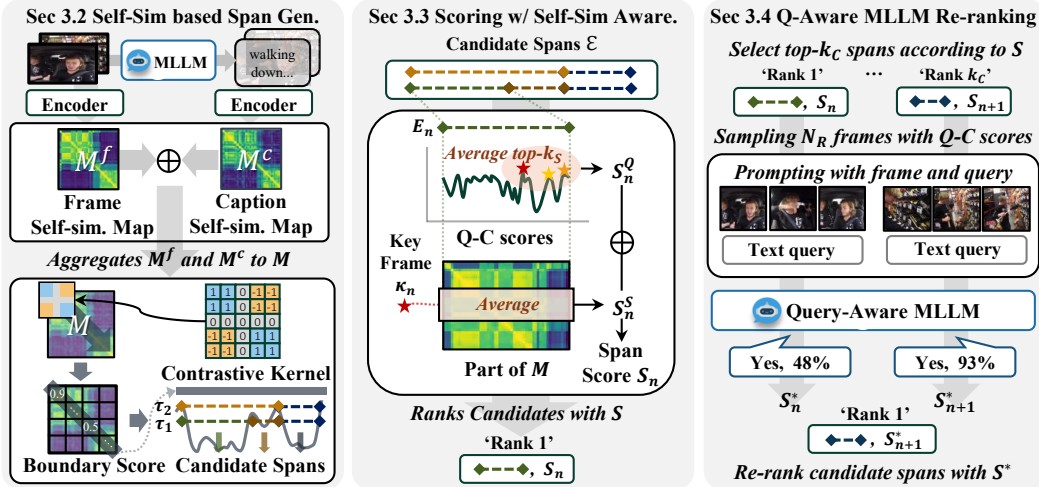

Figure 2: The overview of our Self-Similarity based Moment proposal and Scoring (Self-SiMS). First, features from the frames and captions are extracted to construct two distinct self-similarity maps, $M^f, M^c$, which are then aggregated into a unified map, $M$. Each frame is then evaluated with a boundary score computed using a contrastive kernel and candidate spans are generated in temporal order (Sec. 3.2). Next, the spans are ranked based on query-matching span score $S^Q$, which is computed by averaging the top-$k_S$ query-caption scores. In addition, self-matching span score $S^S$, is derived from the unified Self-similarity map $M$ and then combined with the former score (Sec. 3.3). Finally, the MLLM-based reranking is applied to the top-ranked spans by asking whether each query-frame pair is relevant, and the probability of the 'Yes' answer is used as the score (Sec. 3.4).

where $\mathcal{B}_j$ contains the indices of frames that are both above the threshold and correspond to local maxima of the boundary scores. Here, the two extreme frames, $1$ and $L$, are included in $\mathcal{B}_j$ to ensure that candidate spans cover the entire video. By these boundary sets, we obtain a candidate span $E_n$ defined as:

$$E_n = [e_n^s, e_n^e], \quad 1 \le e_n^s < e_n^e \le L, \tag{3}$$

where each pair $(e_n^s, e_n^e)$ is formed by two successive frame indices in the temporal order within the same boundary set $\mathcal{B}_j$, ensuring that different spans do not overlap. Finally, the set of all candidate spans across thresholds is denoted by

$$\mathcal{E} = \{E_1, E_2, \ldots, E_{N_s}\}, \tag{4}$$

where $N_s$ denotes the total number of candidate spans. Note that, each $E_n$ in this set refers to the corresponding candidate span and can be referred to as the "$n$-th span" in subsequent descriptions.

## 3.3 SPAN SCORING WITH SELF-SIMILARITY AWARENESS

To identify the relevant candidate span, each span should have its relevance score with the query. Previous approach Xu et al. (2025) computes this score by averaging the frame-level similarities between the query and captions. However, such mean-based scoring is unreliable, as query–caption similarity often exhibits noisy and inconsistent distributions due to the language-style gap.

To mitigate this limitation, we use the top-$k_S$ frame-level similarities within the span for scoring, which are robust to the corruption induced by language-style gap. In addition, to enhance the reliability of scoring, we propose Self-Matching Span Score, derived from TSM. Finally, each candidate span is scored by combining the Query-Matching Span Score and the Self-Matching Span Score.

**Query–Matching Span Score** Given a textual query $Q$, we extract the query feature $F^q$ by the encoder used to extract the caption features $F^c$. Then, the cosine similarity between the query feature and caption features is calculated to produce the frame-level scores $S^f \in \mathbb{R}^L$ as follows:

$$S_i^f = \frac{F^q \cdot F_i^c}{\|F^q\| \, \|F_i^c\|}, \tag{5}$$

where $F_i^c$ and $S_i^f$ denote the caption feature and frame-level score, respectively, of the $i$-th frame. The Query-Matching Span Score (QMS), denoted as $S^Q$, is defined as the mean of the top-$k_S$ frame scores within each span. Specifically, for the $n$-th span $E_n$, containing the indices from the start to the end frames $e_n^s$ and $e_n^e$, the score is computed as follows:

$$S_n^Q = \frac{1}{k_S} \sum_{i=e_n^s}^{e_n^e} \mathbf{1}\left[i \in I_{k_S}^{E_n}\right] S_i^f,$$ (6)

where $I_{k_S}^{E_n}$ is a set of indices with the top-$k_S$ highest frame-level scores within the candidate span. This approach prioritizes the most relevant frames over the average, helping to mitigate noise in the query–caption similarity distribution by using the most reliable scores. Consequently, the span score more accurately reflects the true relevance of the query.

**Self-Matching Span Score**   However, relying solely on the top-$k_S$ frame scores constrains the ability to capture the alignment between the query and the entire span context, as the resulting score is derived from only a subset of frames rather than the whole span. To address this issue, we leverage the TSM used in span generation to evaluate the alignment between the query and the entire span context while avoiding the influence of the language-style gap. The Self-Matching Span Score (SMS), denoted as $S^S$, is defined as the mean similarity between the key frame with the highest frame-level score in a span and all other frames within the same span. Let $\kappa_n$ denote the index of the key frame in the span $E_n$, that is,

$$\kappa_n = \arg\max_{i \in E_n} S_i^f.$$ (7)

Then, SMS for the $n$-th candidate span is calculated as

$$S_n^S = \frac{1}{|E_n|} \sum_{j \in E_n} M_{\kappa_n, j},$$ (8)

where $M_{i,j}$ denotes the element of the TSM $M$ at the position $(i, j)$. As a result, candidate spans that are longer yet include inconsistent contexts are suppressed through reduced span scores. Finally, the total score $S_n$ of the candidate span is weighted sum of $S_n^Q$ and $S_n^S$ as follows:

$$S_n = (1 - \alpha) S_n^Q + \alpha S_n^S, \quad \alpha \in [0, 1],$$ (9)

where $\alpha$ is a hyperparameter for the span score weights.

### 3.4 QUERY-AWARE GAP REDUCTION VIA MLLM RE-RANKING

Recent advances in multimodal large language models (MLLMs) show strong cross-modal reasoning ability, making them effective for re-ranking tasks Lin et al. (2024); Chen et al. (2024); Sun et al. (2023a); Jin et al. (2025). Building on this, we mitigate the language-style gap between queries and MLLM-generated captions by refining candidate spans through MLLM-based re-ranking.

Concretely, we re-rank the top-$k_C$ candidate spans from the initial scoring stage, ranked according to their span Score $S$. Within each selected span, we sample $N_R$ representative frames according to their query–caption score. Each frame is evaluated by the MLLM with a *Yes/No* prompt, and the logits are converted into probabilities via softmax; the *Yes* probability serves as the frame-level score $S_i^r$. Span-level re-ranking scores $S^R$ are then obtained by averaging the top-$k_R$ frame-level score $S_i^r$, and finally combined with the original span score $S_n$ through a weighted sum:

$$S_n^* = (1 - \beta) S_n + \beta S_n^R, \quad \beta \in [0, 1],$$ (10)

where $\beta$ is a weighting parameter. $S_n^*$ represents the final re-ranked score for each top-$k_C$ candidate span, integrating both the initial score $S_n$ and the MLLM-based re-ranking signal $S_n^R$. This re-ranking process leverages the MLLM's cross-modal reasoning to directly verify semantic alignment between the query and candidate captions, rather than relying solely on similarity scores. Unlike prior usage of MLLMs that verify captions, we explicitly provide the query together with the caption, enabling query-aware reasoning and more reliable selection of the most relevant spans. Further details are provided in the Appendix.

| Method | Setting | QVHighlights test | | | | QVHighlights val | | | |
|---|---|---|---|---|---|---|---|---|---|
| | | R1@0.5 | R1@0.7 | mAP@0.5 | mAP@avg | R1@0.5 | R1@0.7 | mAP@0.5 | mAP@avg |
| Moment-DETR Lei et al. (2021) | FS | 52.9 | 33.0 | 54.8 | 30.7 | 54.2 | 33.4 | 55.4 | 31.1 |
| UMT Liu et al. (2022b) | FS | 56.4 | 40.8 | 53.1 | 35.4 | - | - | - | - |
| MomentDiff Li et al. (2023b) | FS | - | - | - | - | 57.8 | 39.2 | 54.6 | 35.3 |
| CNM Zheng et al. (2022a) | WS | 14.1 | 4.0 | 11.8 | - | - | - | - | - |
| CPL Zheng et al. (2022b) | WS | 30.8 | 10.8 | 22.8 | - | - | - | - | - |
| CPI Kong et al. (2023) | WS | 32.3 | 11.8 | 23.7 | - | - | - | - | - |
| Liu et al. (2022a) | US | - | - | - | - | 12.3 | 3.5 | 10.4 | 2.7 |
| PZVMR Wang et al. (2022) | US | 14.2 | 4.9 | 15.7 | 4.6 | 12.6 | 5.1 | 16.2 | 5.3 |
| Diwan et al. (2022) | ZS | - | - | - | - | 48.3 | 31.0 | 47.3 | 28.0 |
| Wattasseril et al. (2023) | ZS | 52.4 | 31.6 | 51.7 | 29.6 | 53.1 | 32.2 | 52.3 | 30.2 |
| Moment-GPT Xu et al. (2025) | ZS | 58.3 | 37.7 | 55.1 | 35.0 | 58.9 | 38.6 | 55.7 | 35.9 |
| **Self-SiMS (Ours)** | ZS | **59.7** | **42.2** | **59.2** | **38.3** | **61.0** | **43.2** | **60.5** | **39.3** |

Table 1: Comparison on QVHighlights test and validation sets. FS: Fully Supervised, WS: Weakly Supervised, US: Unsupervised, ZS: Zero-Shot.

| Method | Setting | Charades-STA | | | | ActivityNet-Captions | | | |
|---|---|---|---|---|---|---|---|---|---|
| | | R1@0.3 | R1@0.5 | R1@0.7 | mIoU | R1@0.3 | R1@0.5 | R1@0.7 | mIoU |
| TimeChat Ren et al. (2024) | FS | - | 43.8 | 22.7 | - | - | - | - | - |
| Moment-DETR Lei et al. (2021) | FS | 62.1 | 48.2 | 25.3 | 42.3 | 52.6 | 32.5 | 15.3 | 37.8 |
| CNM Zheng et al. (2022a) | WS | 50.0 | 36.2 | 14.2 | 34.2 | 51.3 | 30.3 | 11.4 | 33.9 |
| CPL Zheng et al. (2022b) | WS | 56.0 | 38.1 | 20.3 | 37.8 | 52.4 | 30.9 | 12.0 | 32.6 |
| Huang et al. (2023) | WS | 59.2 | 44.2 | 22.1 | 39.4 | 54.8 | 32.9 | - | 36.4 |
| PSVL Nam et al. (2021) | US | 45.2 | 30.9 | 14.2 | 30.9 | 45.1 | 29.8 | 15.7 | 30.2 |
| Gao & Xu (2021) | US | 45.3 | 19.8 | 7.9 | - | 45.8 | 25.9 | 12.1 | - |
| Liu et al. (2022a) | US | 44.2 | 28.7 | 14.7 | - | 47.3 | 28.2 | - | - |
| TFVTG† Zheng et al. (2024) | ZS | **64.7** | 29.8 | 9.8 | 38.2 | 49.9 | 26.8 | 12.6 | 34.2 |
| Moment-GPT Xu et al. (2025) | ZS | 58.2 | 38.4 | **21.6** | 36.5 | 48.1 | **31.1** | **14.9** | 30.8 |
| **Self-SiMS (Ours)** | ZS | 62.7 | **39.7** | 21.0 | **41.9** | 49.9 | 28.2 | 13.8 | **34.7** |

Table 2: Comparative evaluation on Charades-STA and ActivityNet-Captions. FS: Fully Supervised, WS: Weakly Supervised, US: Unsupervised, ZS: Zero-Shot. †: Reproduced results.

## 4 EXPERIMENTS

### 4.1 EXPERIMENTAL SETUP

**Datasets** We evaluate our method on three widely used benchmarks. QVHighlights Lei et al. (2021) contains diverse user-generated videos annotated with subtle, highlight-level moments. Charades-STA Gao et al. (2017) consists of indoor daily activities paired with temporal sentence annotations that describe specific actions. ActivityNet-Captions Krishna et al. (2017) covers a broad range of human activities with detailed descriptive captions aligned to temporal segments, providing a challenging testbed for video moment retrieval (VMR).

**Metrics** Following prior zero-shot video moment retrieval (ZMR) works, we evaluate our method using standard metrics tailored to each dataset. For QVHighlights, we report Recall@1 at IoU thresholds of 0.5, and 0.7, along with mAP@0.5 and the average mAP across multiple thresholds. These metrics jointly reflect the retrieval accuracy and the quality of segment ranking. For Charades-STA and ActivityNet-Captions, we adopt R1@0.3/0.5/0.7 and mean IoU (mIoU), which assess the precision of temporal localization as well as the alignment between predicted and ground-truth segments.

**Implementation details** Following prior work Xu et al. (2025), we set the frame rates of QVHighlights, Charades-STA, and ActivityNet-Captions to 0.5, 1, and 1, respectively. We employ LLaMA-3.2-11B-Vision-Instruct Grattafiori et al. (2024) for both video frame captioning and the final re-ranking stage. For scoring, the number of top scores is fixed at 3 for both initial span scoring ($k_S$) and re-ranking ($k_R$). We re-rank the top-$k_C$ candidate spans with $k_C = 5$. In the re-ranking stage, $N_R$ frames are sampled from each span, set to half of the span length with lower and upper bounds of 10 and 30, respectively. The weighting parameters for the final span score, $\alpha$ and $\beta$, are set to 0.1 and 0.5, respectively. All experiments are conducted on an NVIDIA Quadro A6000 GPU.

### 4.2 COMPARISON WITH THE STATE-OF-THE-ARTS

Tab. 1 and Tab. 2 report the performance of VMR methods. In Tab. 1, Self-SiMS outperforms existing ZMR baselines across all evaluation metrics on QVHighlights. Tab. 2 presents the performance comparison on Charades-STA and ActivityNet-Captions. To ensure fairness, we reproduce

| Method | Oracle-mIoU (%) | Avg. # Span Cand. |
|---|---|---|
| TFVTG | 48.24 | 7.05 |
| Moment-GPT | 48.49 | 4.41 |
| Self-SiMS (Ours) | **57.72** | **3.22** |

Table 3: Comparison of Oracle mIoU on the Charades-STA test set. Avg. # Span Cand. denotes the average number of span candidates across on test set.

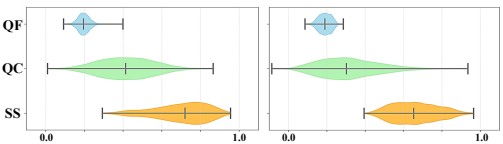

Figure 3: Similarity distributions within ground-truth moments on the QVHighlights (left) and Charades (right), where QF, QC, and SS denote query–visual frame, query–caption, and self-similarity.

| Method | mAP | mAP@0.5 | R1@0.5 | R1@0.7 |
|---|---|---|---|---|
| Mean scoring | 37.10 | 56.69 | 55.74 | 40.90 |
| QMS | 38.63 | 59.41 | 59.48 | 42.13 |
| + SMS | 38.72 | 59.59 | 60.26 | 42.90 |
| + Re-ranking | **39.27** | **60.51** | **60.97** | **43.16** |

Table 4: Scoring component analysis of our scoring methods. QMS denotes the Query-Matching Score, and SMS denotes the Self-Matching Span Score.

| Method | mAP | mAP@0.5 | R1@0.5 | R1@0.7 |
|---|---|---|---|---|
| Baseline | 38.72 | 59.59 | 60.26 | 42.9 |
| + Rephrasing | 38.50 | 59.04 | 59.55 | 42.90 |
| + Re-ranking | **39.27** | **60.51** | **60.97** | **43.16** |

Table 5: Comparison between our MLLM re-ranking and the query rephrasing technique. The Baseline denotes the performance of our model without MLLM re-ranking.

TFVTG[†] Zheng et al. (2024) at 1 fps, as the original results were reported at 0.5 and 3 fps. TFVTG attains relatively high R@0.3 and mIoU but shows clear drops on R1@0.5 and R1@0.7, as its reliance on query–visual feature similarity leads to long spans and reduced recall at stricter thresholds. In contrast, Moment-GPT achieves stronger R1@0.5 and R1@0.7 but lower R1@0.3 and mIoU, reflecting the unstable nature of query–caption similarity, which can produce precise short matches in some cases yet frequent failures in others. This contrast highlights the different sensitivities of evaluation metrics: long spans may inflate recall at relatively low IoU thresholds (e.g., 0.3) while lowering mIoU, whereas unstable short predictions yield the opposite. In contrast, our Self-SiMS overcomes these issues by producing tighter and more consistent spans aligned with the ground-truth, achieving the highest mIoU and more reliable temporal localization while maintaining competitive recall. These results demonstrate that Self-SiMS delivers not only balanced performance across different metrics but also robust temporal localization under diverse evaluation settings.

### 4.3 ABLATION STUDIES

**Effectiveness of Span Generator**    To validate the effectiveness of our span generator, we evaluate Oracle mIoU on the Charades-STA, comparing against TFVTG Zheng et al. (2024) and Moment-GPT Xu et al. (2025). Oracle mIoU is defined as the mean IoU obtained by selecting the candidate span with the maximum overlap with the ground truth, reflecting the quality of the span candidates. As shown in Tab. 3, our method achieves an Oracle mIoU of 57.72, outperforming both baselines. Moreover, although this performance generally increases with a larger number of candidate spans, our method achieves the best results despite requiring the lowest average number of candidates (3.22). This improvement demonstrates that our method generates more reliable spans by capturing the intrinsic structure of the video while avoiding modality and language-style gaps.

**Score Distribution in Ground-Truth Moments**    To study the impact of modality and language-style gaps on span generation, we analyze similarity distributions within ground-truth moments on the QVHightlights validation set and Charades-STA. Specifically, we compare query–visual frame similarity (QF) in TFVTG Zheng et al. (2024), query–caption similarity (QC) in Moment-GPT Xu et al. (2025), and our proposed self-similarity (SS), the mean of similarities within the ground-truth span. As shown in Fig. 3, QF yields low and narrowly concentrated scores due to the modality gap between queries and frames. QC exhibits a wider distribution, reflecting style differences between human queries and MLLM-generated captions. By contrast, SS produces consistently higher and more stable values across both datasets, highlighting its ability to generate reliable span candidates.

**Scoring Component Analysis**    We conduct an ablation study on the QVHighlights validation set to analyze the contribution of each component in our proposed scoring and re-ranking pipeline, with the results presented in Tab. 4. The baseline adopts naive mean scoring, which averages all frame-level

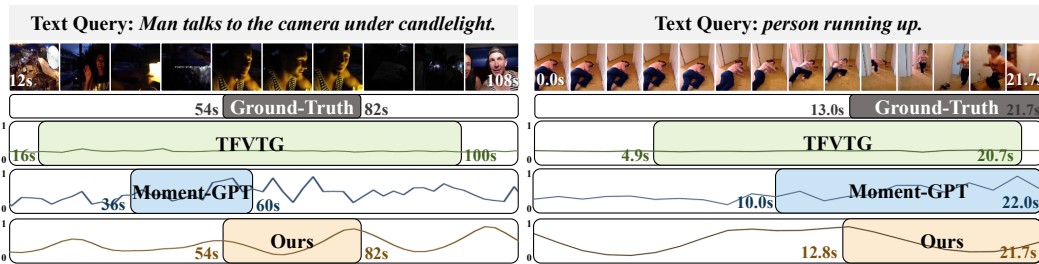

Figure 4: Qualitative results on QVHighlights (left) and Charades-STA (right).

scores within a span, as in prior methods. Replacing mean scoring with the Query-Matching Span Score (QMS) yields improvements across all metrics, supporting our hypothesis that employing the most relevant signals is more effective than relying on noisy mean scoring. Adding the Self-Matching Span Score (SMS) on QMS further increases performance, particularly for recall-based metrics such as R1@0.5 and R1@0.7, indicating that incorporating the entire span context through self-similarity enables more precise localization of ground-truth moments. Finally, incorporating MLLM re-ranking on top of QMS with SMS yields additional gains across all metrics by leveraging the MLLM's ability to understand cross-modal information. This demonstrates that each component of the pipeline contributes effectively to the overall performance.

**Strategy Comparison for Language-Style Gap Mitigation**    To validate the effectiveness of our query-aware gap reduction via MLLM re-ranking, we compare our approach against query rephrasing, which has been employed in prior work Zheng et al. (2024); Xu et al. (2025). Query rephrasing reformulates a query into semantically equivalent variants with alternative wordings, thereby promoting query generalization and mitigating the language gap. To provide a fair comparison, we first consider our method without MLLM re-ranking as the baseline. We then compare it with a variant that incorporates Moment-GPT's query rephrasing for re-ranking and with our full method using MLLM re-ranking. As shown in Tab. 5, our query-aware MLLM re-ranking achieves clear and consistent improvements across all metrics compared with query rephrasing. These results demonstrate that leveraging MLLM-based query-aware reasoning more effectively bridges the language-style gap, resulting in more accurate scoring and improved moment retrieval performance.

### 4.4    QUALITATIVE RESULTS

We provide a qualitative comparison in Fig. 4, illustrating span localization results, TFVTG Zheng et al. (2024), Moment-GPT Xu et al. (2025), and our method, along with the corresponding span candidate score distributions. TFVTG, which relies on query–visual frame similarity, generates overly long and inaccurate spans due to the modality gap that hinders fine-grained semantic alignment. Moment-GPT, which instead depends on query–caption similarity, suffers from a language-style gap that produces inconsistent similarity maps and hampers accurate localization. In contrast, Self-SiMS leverages boundary scores derived from intra-video self-similarity, enabling robust localization of the correct spans while effectively mitigating both the modality and language-style gaps.

## 5    CONCLUSION

In this paper, we propose Self-Similarity based Moment proposal and Scoring (Self-SiMS), a novel framework based on self-relationships within video for zero-shot video moment retrieval (ZMR), to circumvent modality and language-style gaps. The proposed method leverages Temporal Self-Similarity Matrix not only for span generation but also for scoring, thereby composing contextually consistent candidate spans and mitigating errors from noisy query-caption similarity scores. Experimental results demonstrate that our approach outperforms the baseline methods that suffer from inaccurate span generation caused by modality and language-style gaps. Furthermore, the ablation studies confirm that our span generation method produces more accurate spans and re-ranking method mitigates the language-style gap effectively. In conclusion, our Self-SiMS suggests that self-similarity can be considered a useful cue in ZMR.

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

# A APPENDIX

## A.1 DEFINITION OF THRESHOLD FOR GENERATING CANDIDATE SPANS

In this section, we define the instance-wise dynamic threshold $\tau_j$, which suppresses the frames with low boundary scores. The motivation for adopting a dynamic threshold is that segmenting videos into contextually consistent spans is highly sensitive to the variability of the context within the video. A fixed threshold, which ignores this factor, fails to segment the videos appropriately. Therefore, we propose the set of thresholds $\mathcal{T} = \{\tau_1, \tau_2, \ldots, \tau_{N_{\mathcal{T}}}\}$ which adapt to the video context variability.

**Video Context Variability** We introduce an indicator of intra-video context variability to make the threshold respond to it. First, we $\ell_2$-normalize the frame features $F^f \in \mathbb{R}^{L \times D^f}$ into $\hat{F}^f$. Then, we obtain the mean of the normalized frame features $\mu^f$ as follows:

$$\mu^f = \frac{1}{L} \sum_{i=1}^{L} \hat{F}_i^f. \tag{11}$$

Then, we apply zero-centering on $\hat{F}_i^f$, that is, $\tilde{F}_i^f = \hat{F}_i^f - \mu^f$. We define the variability of the video context $\mathcal{V}$ by averaging the $\ell_2$-norm of zero-centered $\tilde{F}_i^f$:

$$\mathcal{V} = \frac{1}{L} \sum_{i=1}^{L} \|\tilde{F}_i^f\|_2. \tag{12}$$

The above formulations can be interpreted as follows: if the original frame features $F^f$ are uniformly spread over an arbitrary hypersphere, the mean vector $\mu^f$ computed after $\ell_2$-normalization will be close to the zero vector, which in turn makes $\mathcal{V}$ close to 1. Conversely, if the frame features are biased toward a particular direction, which means that some frame features have similar context, the mean vector $\mu^f$ will also be shifted toward that direction, resulting in a smaller norm of $\tilde{F}^f$ and consequently a lower $\mathcal{V}$. In other words, $\mathcal{V}$ reflects the diversity of the contexts contained in a video instance. Therefore, we regard $\mathcal{V}$ as an indicator of the intra-video context variability.

**Definition of Threshold** Next, we define the threshold $\tau_j$ based on the video context variability. It is obvious that videos with diverse contexts should undergo fine-grained segmentation whereas homogeneous ones require coarse-grained segmentation. Thus, the threshold should decrease as the indicator $\mathcal{V}$ grows, that is, leading to more candidate spans by increasing the size of boundary set $\mathcal{B}_j$. In contrast, the threshold should grow as the indicator $\mathcal{V}$ decreases, thereby reducing the number of candidate spans by suppressing more frames. Therefore, we specify the threshold as inversely related to the variability:

$$\tau_j = 1 - \lambda_j \mathcal{V} \quad \text{where } \lambda_j \in \Lambda_{j=1}^{N_\tau}, \tag{13}$$

where $\Lambda$ is a set of constants that control threshold scaling and the generation of spans with diverse lengths. In practice, we set $\Lambda$ to $\{1, 2, 3\}$ across the benchmark datasets, hence $N_\tau = 3$. Note that the threshold changes instance-wise via $\mathcal{V}$.

## A.2 DETAILED PROCEDURE OF MLLM-BASED RE-RANKING

In this section, we provide a more detailed explanation of the MLLM-based re-ranking process, which complements the concise description in the main paper.

**Candidate Selection** For efficiency, re-ranking is applied only to the top-$k_C$ candidate spans from the initial scoring stage, as these are the most likely to contain the correct answer. In our implementation, we set $k_C = 5$, balancing computational efficiency with sufficient coverage of promising spans. The selected candidate spans are represented as

$$\mathcal{E}_R = \{[e_n^s, e_n^e]\}_{n=1}^{k_C}, \tag{14}$$

where $e_n^s$ and $e_n^e$ denote the start and end frame indices of the $n$-th selected span, forming the set of spans subject to re-ranking. For the $n$-th span $[e_n^s, e_n^e]$, the number of frames is defined as

$$N_n^f = e_n^e - e_n^s + 1. \tag{15}$$

**Frame Sampling.** For each span, our goal is to extract representative frames that best capture the semantic content relevant to the query. To achieve this, we use the previously computed query–caption (QC) scores $S^f$, which measure the alignment between the query and individual video frames. Frames with higher QC scores are prioritized for sampling, as they are assumed to be more informative for query verification. The number of sampled frames is determined adaptively according to the span length with a sampling rate $\rho$, and is clamped between a minimum and maximum to avoid undersampling short spans or oversampling long spans:

$$N_R = \max\big(N_{\min}^f, \min(\rho \cdot N_n^f, N_{\max}^f)\big), \tag{16}$$

where $N_{\min}^f$ and $N_{\max}^f$ are 10 and 30, respectively. The $N_R$ frames with the highest QC scores are retained, and their indices are sorted in temporal order to preserve sequential structure.

**Prompting and Scoring** The sampled frames are passed to the MLLM for comparison with the text query. To ensure consistent behavior, we employ a structured prompt that asks the model whether the query is aligned with the input frame. The prompt is designed as:

> *"You are a visual fact-checker. Your job is to verify a statement against a single video frame. Do not infer or guess information that is not clearly visible. Statement to verify:* **[Query]**. *Based strictly on the visual information in the image, is this statement true? Answer with only 'Yes' or 'No'."*

Here, **[Query]** denotes the placeholder where the text query is inserted. The MLLM produces logits for both "Yes" and "No." Let $l_i^{\text{yes}}$ and $l_i^{\text{no}}$ denote the logits corresponding to $i$-th frame. The frame-level re-ranking score is defined as the softmax probability of the "Yes" token:

$$S_i^r = \frac{\exp(l_i^{\text{yes}})}{\exp(l_i^{\text{yes}}) + \exp(l_i^{\text{no}})} \tag{17}$$

**Span-Level Score Aggregation** To obtain a re-ranking score for an entire span, we aggregate the frame-level scores. Instead of averaging all sampled frames, which may include noisy or less informative ones, we select the top-$k_R$ verification scores (with $k_R = 3$) and compute their mean:

$$S_n^R = \frac{1}{k_R} \sum_{i=e_n^{\text{s}}}^{e_n^{\text{e}}} \mathbf{1}\left[i \in I_{k_R}^{E_n}\right] S_i^r, \tag{18}$$

where $I_{k_R}^{E_n}$ is a set of indices with the top-$k_R$ highest frame-level re-ranking scores within the $n$-th candidate span $E_n$. This design emphasizes the most confident evidence while reducing the impact of outliers.

### A.3 SCOPE OF LLM USAGE

In accordance with Policy 1 of ICLR's guidelines on the use of large language models (LLMs), we disclose that LLMs were used solely for refining grammar and improving the clarity of expression during paper writing. All authors have thoroughly reviewed the content drafted with LLMs to ensure that it contains no falsehood, plagiarism, or misrepresentation. In line with Policy 2, the authors take full responsibility for the contents of this submission.

