# OpenReview forum: "Mitigating Modality and Language-Style Gaps for Zero-Shot Video Moment Retrieval"
_ICLR.cc/2026/Conference — ICLR 2026 Conference Withdrawn Submission_

### Official Review · Reviewer_X4J3 · 2025-10-27

**Soundness:** 3
**Presentation:** 3
**Contribution:** 2
**Rating:** 4
**Confidence:** 4

**Summary:**

This paper addresses zero-shot video moment retrieval. The method proposes Self-Similarity-based Moment Proposal and Scoring (Self-SiMS), which overcomes the unreliable similarity signals in previous zero-shot video moment retrieval methods. With the proposed method, final predictions are obtained by multimodal large language models with a tailored prompt. Self-SiMS demonstrates superior performance across datasets and models under various supervision levels.

**Strengths:**

1. Self-SiMS sophisticatedly constructs similarity scores between video, query, and caption. This directly leads to accurate grounding results, supporting its effectiveness.

2. Self-SiMS achieves competitive and strong performance across benchmarks, outperforming previous models under various supervision levels.

3. The ablation studies fairly explore the role of components in the method.

**Weaknesses:**

- I think the method does not explicitly bridge the gap between visual and textual representations, but rather provides a more sophisticated way to measure distances between frozen modalities, as it does not involve any training process. Consequently, its effectiveness may depend heavily on the strength of the underlying encoders, which is not discussed or explored (only LLaMA-3.2-11B-Vision-Instruct is used). Since different encoders capture distinct visual signals in videos [1]. For example, GroundingDINO [3] is likely better at object-centric representations. Furthermore, VMR performance is highly sensitive to the choice of visual features; models using I3D or CLIP+SlowFast features often outperform those based on VGG, as widely reported in prior studies [4, 5]. Exploring how encoder choices influence similarity scores could therefore provide valuable insights.

- There is no ablation study for the hyperparameters $\alpha$ and $\beta$. These parameters balance which similarity is more important for predictions and directly impact performance. For example, why does $\alpha$=0.1 work and lead to good performance? In this case, the model rarely references the Self-Matching Span Score during predictions. Could the authors elaborate on this?

- The process during MLLM-ReRanking resembles the verification probes in [2], which ask a model to verify whether a given event happens in a specific video moment. [2] showed that the verification capabilities in MLLMs are not actually high and might introduce noise. Have the authors considered the potential noise in MLLMs during reranking?

- It is natural that a model needs to see and sample more frames for accurate predictions. How's the efficiency of the method and the impact of frame numbers? Could authors compare the efficiency between zero-shot methods and Self-SiMS?

- (Minor) The paper already demonstrates the effectiveness of Self-SiMS through quantitative results. Including additional qualitative analyses (e.g., a visual comparison of similarity scores and corresponding predictions across methods) could further enhance the paper. Such examples would provide deeper insight into how Self-SiMS captures cross-modal similarities, making the contribution more intuitive beyond standard success or failure cases.

**References**

[1] Unifying Specialized Visual Encoders for Video Language Models, ICML 2025

[2] On the Consistency of Video Large Language Models in Temporal Comprehension, CVPR 2025

[3] Grounding DINO: Marrying DINO with Grounded Pre-Training for Open-Set Object Detection, ECCV 2024

[4] Background-aware Moment Detection for Video Moment Retrieval, WACV 2025

[5] Unified Static and Dynamic Network: Efficient Temporal Filtering for Video Grounding, TPAMI 2025

**Questions:**

I'm open to revising the rating if the authors address the weaknesses.

- I'm not sure if Fig. 1-(a) (Ours) represents that the proposed method truly avoids the gap between modalities. It feels like showing the difference between methods, rather than suggesting that the method overcomes the issue.

- I think additional discussion on related weakly supervised [1, 2] and unsupervised [3, 4] VMR works could be helpful. But this is not essential, as I acknowledge that the focus of this paper is distinct from these works.

**References**

[1] Weakly Supervised Temporal Sentence Grounding with Gaussian-based Contrastive Proposal Learning, CVPR 2022

[2] Iterative Proposal Refinement for Weakly-Supervised Video Grounding, CVPR 2023

[3] Modal-specific Pseudo Query Generation for Video Corpus Moment Retrieval, EMNLP 2022

[4] Language-free Training for Zero-shot Video Grounding, WACV 2023

---

### Official Review · Reviewer_4AsS · 2025-10-28

**Soundness:** 3
**Presentation:** 2
**Contribution:** 3
**Rating:** 4
**Confidence:** 4

**Summary:**

This paper proposes Self-SiMS, a zero-shot video moment retrieval framework that aims to mitigate modality and language-style gaps. The method leverages intra-video self-similarity to generate moment candidates and combines it with a carefully designed span scoring mechanism that integrates both query–caption relevance and intra-span consistency. A query-aware MLLM re-ranking stage further refines the final predictions. The paper is clearly written, and experiments show consistent improvements over strong baselines such as Moment-GPT and TFVTG.

**Strengths:**

1. The combination of query–caption similarity (top-k aggregation) and intra-span self-similarity provides a balanced and stable scoring signal that effectively mitigates noise from cross-modal misalignment.
2. The proposed method achieves state-of-the-art zero-shot results across multiple benchmarks, demonstrating both robustness and generalization.
3. Using self-similarity maps for span proposal is a simple but elegant idea that improves reliability by leveraging video-intrinsic structure.
4. Clear and well-organized presentation: Figures and mathematical formulations are easy to follow, and the overall writing quality is strong.

**Weaknesses:**

1. Inaccurate claim of “bypassing” cross-modal similarity:
The paper repeatedly claims that Self-SiMS “circumvents” or “eliminates” cross-modal comparison. In reality, only the proposal span generation stage avoids direct query–context similarity, while the scoring stage still relies on query–caption similarity. Thus, the method stabilizes rather than removes cross-modal dependence.

2. Lack of deeper analysis:
The paper could provide a stronger justification for why self-similarity correlates with semantic boundaries and examine how sensitive the performance is to hyperparameters (e.g., α, β, kernel size).

3. Limited contribution of the Self-Matching Span Score:
As shown in Table 4, the Self-Matching Span Score has limited impact, while the reranking module contributes most of the performance gains. This observation contradicts the paper’s central claim.

4. Missing ablation on segmentation strategies:
The paper lacks ablation studies that explore alternative video segmentation methods followed by the MLLM-based reranker.

5. Scope of claims:
Certain statements in the introduction and abstract overstate the theoretical generality of the method; adopting more precise wording would strengthen the paper’s credibility.

**Questions:**

Same as the Weaknesses.

---

### Official Review · Reviewer_hLgp · 2025-10-31

**Soundness:** 3
**Presentation:** 3
**Contribution:** 2
**Rating:** 4
**Confidence:** 3

**Summary:**

This paper presents a zero-shot video moment retrieval method. It specifically addresses the dependence of existing zero-shot methods on query-to-context similarity which suffer from modality and language-style gaps. To overcome this, the paper proposes Self-Similarity-based Moment proposal and Scoring to generate consistent candidates. It also introduces a query-aware MLLM-based reasoning stage to further refine the alignment between text and video. SOTA results are reported on multiple zero-shot video moment retrieval benchmarks.

**Strengths:**

The paper is well written. The structure is easy to follow, and illustrations are clear.

The idea of self-similarity to address the modality gap in zero-shot video moment retrieval is interesting and plausible.

Boundary scores calculation to generate candidate spans from self-similarity of frame/caption features is likely to work well.

Besides improvement of the event boundary detection, adding caption information in addition to the visual features is an interesting direction.

The idea of using MLLM to re-rank the candidate spans is also interesting and should lead to improved performance.

The overall method seems to be well designed and convincing.

**Weaknesses:**

The idea of self-similarity to address modality gap is not new. If fact, it is one of the earliest methods I can recall in this line of work. I am not sure whether it has been used in the context of video-moment retrieval or not. Never-the-less, the novelty of the proposed method remains low as this would be just another application. I am keen so see how the other reviewers perceive this and remain open to being convinced otherwise during the rebuttal period.

Vision-language models such as CLIP are extensively trained to reduce the modality gap. How is self-similarity better?

In the Introduction, the idea of event boundaries is linked with shot/scene clustering only. However, later on in the methods, information about additionally using captions is presented. For completeness, this information should also be mentioned in the Introduction, as it is a vital part of the method.

The candidate spans will vary significantly based on the chosen threshold. Did you do an experiment for this?

There are other adhoc choices and hyper-parameters e.g. top-$k_S$ frame-level scores, SMS calculation, and $\alpha$, $\beta$ hyper-parameters. How sensitive is the method to these choices?

The framework's ability to be scaled for practical settings remains unclear. The added complexity and costs seem to outweigh the performance gains compared to existing works. For example, the gains on R1@0.5 metric over existing works are not that high in general and much low in the case of ActivityNet-Captions dataset.

The ablations in Table 4 show minimal improvement provided by adding SMS and Re-ranking modules. Similarly, in Table 5, Baseline is as almost as good as MLLM Re-ranking. The significant overhead of using an MLLM does not seem justified.

Typo on page 4: “Then, The TSMs ...” -> Then, the TSMs...

**Questions:**

See the Weaknesses section.

---

### Official Review · Reviewer_mgRe · 2025-10-31

**Soundness:** 3
**Presentation:** 3
**Contribution:** 3
**Rating:** 4
**Confidence:** 3

**Summary:**

This paper proposes a zero-shot video moment retrieval framework aiming to mitigate the modality and language-style gaps.
Instead of relying solely on query-to-context similarity, they build self-similarity matrices within the video and uses them for span generation and scoring, followed by a query-aware MLLM re-ranking step.
Experiments across QVHighlights, Charades-STA, and ActivityNet-Captions show state-of-the-art zero-shot results.

**Strengths:**

1, The paper tries to solve an important problem for zero-shot vmr.
2, Carry extensive experiments
3，Shows clear performance improvement on one of the dataset
4, the method is simple and easy to plug into different methods.

**Weaknesses:**

1, The experiments seems only works on Qvhighlight, but not on the other two datasets
2, The idea of using self-similarity to generate/assist with boundary  score is not new and has been explored. For example in PZVMR.
3,  Not clear about the main problem they try to solve, multi-modality gap. Would be better if provide with more examples and whats the score after the solution.

**Questions:**

1， What multi-modality,language-style gap mentioned a couple of times in this paper, especially mentioned in l113: mismatches between text queries and video contexts.
2, What is the computational cost for this?

---

### Note · Authors · 2025-11-13

I have read and agree with the venue's withdrawal policy on behalf of myself and my co-authors.